# Numerical Study on Cyclic Response of End-Plate Biaxial Moment Connection in Box Columns

**Marco Gallegos [1], Eduardo Nuñez [2,*] and Ricardo Herrera [3,*]**

[1] Dept. of Structural and Geotechnical Engineering, Pontificia Universidad Católica de Chile, Santiago de Chile, Chile; mfgallegos@uc.cl
[2] Dept. of Civil Engineering, Universidad Católica de la Santísima Concepción, Concepción, Chile
[3] Dept. of Civil Engineering, Universidad de Chile, Santiago de Chile, Chile
**\*** Correspondence: enunez@ucsc.cl (E.N.); riherrer@ing.uchile.cl (R.H.); Tel.:+56-9-51277382 (E.N.)

**Abstract:** The 2008 Wenchuan-China earthquake showed the importance of considering the bidirectional seismic action as a cause of failure in column hinge mechanisms. Subsequently, the large 2011 Tohoku-Japan earthquake revealed that Special Moment Frames buildings, made of tubular columns (Hollow Structural Section or Built-up Box Section) and rigid connections with I-beams, did not suffer serious damage. However, only the ConXtech® ConXL™ moment connection has been prequalified according to the (American Institute of Construction) AISC Seismic Provisions for use with tubular columns and the rest of connections do not consider biaxial resistance. The research reported herein investigated the cyclic response of box-columns joints, connected to I beams using the four-bolt extended endplate connection, subjected to bidirectional bending and axial load on the column. To conduct the study, complex nonlinear finite element models (FEMs) of several I beam to box column joint configurations were constructed and analyzed under cyclic loading using the ANSYS software. The results reveal that the failure is concentrated in the beams of all joint configurations except for the columns with axial load equal to 75% of the column capacity, where a combined failure mechanism is achieved. The energy dissipation capacity of joints with a greater number of beams is lower than joints with fewer beams. The bidirectional effect of the seismic action and the level of axial load must be considered to avoid the formation of a column-hinge fragile failure mechanism also the behavior exhibited by 3D joints is more realistic than 2D joints according to real structures.

**Keywords:** bidirectional loading; performance; bolted connection; end-plate connection; moment connections; finite element method; steel structure; seismic design.

## 1. Introduction

The 2008 Wenchuan China Earthquake evidenced the importance to consider bidirectional seismic action in the performance of columns and its incidence in the failure mechanism [1]. The 2011 Great East Japan Earthquake of Mw 9.0 magnitude followed by a tsunami reached a peak ground acceleration (PGA) of 2.7g, with more than 160 aftershocks Mw 5.5 magnitude and higher and generated numerous human and economic losses. Among the structures that showed no collapse or structural damage were those that used tubular columns ("Box section", made from steel plates or "HSS"—hollow structural section) in buildings with steel moment frames, which are widely employed as part of Seismic Moment Resisting Frame buildings (SMRF) in Japan, according to [2,3]. There are several advantages associated with the tubular section as opposed to shapes with open profiles, which are mentioned as follows: i) since the moment of inertia is the same about any axis for round and square tubes, these sections are the most efficient for columns that have the same end restraints in any direction. For different end restraints about the principle axes, a rectangular tube

can be selected with proportions that provide the same column slenderness ratio about the major and minor axes, thereby providing the most efficient use of material, ii) the torsional stiffness of the closed shape and the high weak axis moment of inertia minimize the requirements for lateral bracing of tubular beams and iii) significant ductility for post-Northridge connections is sometimes provided by panel zone yield deformation, but excess panel zone deformation causes a potential for early connection fracture due to excess local inelastic deformation. Box columns have two "webs" which are effective in resisting panel zone shear, and the "web" thickness is usually the same as the flange thickness. As a consequence, panel zone yielding is less likely to occur, and doubler plates are less likely to be required with box columns, according to [4,5]. Finally, HSS shapes provide similar column strength on both principal axes, while having minimal impact on architectural layout, and the shapes are particularly efficient and economical under compression loads.

American seismic specifications [6] require that connections in special steel moment resisting frames be prequalified to be used. Only one connection, the ConXtech® ConXL™, appears as prequalified in [7] for use with I-beam to tubular column joints. The ConXtech® ConXL™ moment connection was tested and patented by [8]. In this connection, the I-beams are connected to HSS or built-up box columns by means of a field-bolted proprietary collar assembly. The prequalification of the connection was performed with reduced flanges beam and concrete filled columns. The width of the column is limited to 400 mm. The goal of this connection was industrialization and elimination of any welding in worksite. Out of 17 cyclic tests performed, only five were tested under biaxial bending (all interior joints with four beams attached to the column). Exterior joints with beams in three sides and corner joints were not studied. Furthermore, the tests were conducted under constant axial load in the column.

A numerical study on the response under biaxial loading of the connection was conducted by [9] using the finite element method and considering 3D joints and different axial load levels (0.2, 0.4, 0.6 and 0.8 Py, where Py = FyAg, Fy is yield stress and Ag is gross area of section). The results obtained showed a ductile failure mechanism according to the seismic provisions [6] philosophy. Additionally, several joint configurations without concrete infill in the column showed hinges in beams when the axial load level in the column was less than 0.4 the capacity of the column. However, a column hinge mechanism was deemed possible with an increase in the axial load level (0.6 and 0.8 Py).

A recent research conducted by [10], performed an analytical, numerical and experimental study of and end-plate moment connection between a wide flange beam and a hollow structural section column (EP-HSS). This connection also eliminated field welding, using outer diaphragms and end-plates bolted in the field. The results showed that the EP-HSS connection was able to reach 5% drift, allowing the development of inelastic action in the beam with column and connection elements remaining elastic, according to the prequalification protocols established in the [6]. However, it must be noted that from a seismic design standpoint it is not recommended to allow for such large drifts to develop in buildings, since even if the structure can withstand the deformation there will be significant nonstructural damage and the story stability will likely be compromised due to second order effects, particularly for high axial loads in the columns. The effect of bidirectional loading combined with axial load was not studied.

A similar connection to Double Split Tee (DST) using hot-rolled shapes was studied by [11]. This connection is an alternative that allows using built-up T-stubs instead of hot-rolled T-stubs. The experimental research showed an acceptable performance according to AISC seismic provisions. The results indicated that the connection can reach 4% drift without strength degradation, keeping plastic deformation within the beam protected zone. A new moment connection with bolted T-stub connection and reduced beam section was studied by [12]. The results obtained show that flexural strength and dissipated energy is similar to that computed for a T-stub connection and higher than values reported for an RBS (Reduce Beam Section) moment connection.

An experimental research involving the testing of ten full-scale moment resisting connections with wide flange beam to square concrete filled steel tube column under simulated seismic loading conditions was performed by [13]. The results of the study indicate that moment resisting connection can be designed for more than 0.045 rad of inelastic story drift to develop under cyclic loading. The

use of interior diaphragms in the column is shown to locally stiffen the joint, but also lead to strain concentrations and fracture of the beam flanges.

A research conducted by [14] on the seismic performance of a composite moment resisting frame comprised of concrete filled tube columns and wide flange beams was investigated experimentally. Results from the tests indicated that the structural performance under the simulated seismic loading was consistent with the expected performance for all three earthquake levels, obtaining an effective seismic performance of composite moment resisting frames with CFT columns.

A new connection with reduced beam section connection, named as Tubular Web RBS connection (TW-RBS) was proposed by [15]. The obtained results indicated that the connection reduces contribution of the beam to the moment strength creating a ductile fuse far from components of the beam-to-column connection. A numerical study was performed by [16] to solve inaccessibility to the inside of box-columns for assembling the continuity plates. The results showed that connections with short stub beam enhance the behavior of connection and the continuity plates can be omitted and AISC seismic provisions requirements being satisfied. In [17], a study conducted employing numerical models calibrated with data from testing between wide flange beam to box section column connection with welded plates and the influence of complete joint penetration welds was performed. The results obtained showed acceptable performance in those connections that use field welded CJP (Complete Joint Penetration) welds. Field welding is discouraged in many countries, particularly under cyclic or seismic loading.

The finite element method (FEM) was used by [18] to evaluate the effects of column axial load, beam lateral support, and number of stiffeners on concrete filled steel tubular (CFT) column to open beam connections. It was shown that external T-stiffeners combined with internal shear stiffeners enhance the hysteretic performance of CFT columns to open beam connections. The SidePlate moment connection has been studied by [19]. The results showed that this type of connection had strength, stiffness and ductility comparable to a rigid, full-strength and ductile connection. Additionally, [20] studied the performance of HSS-to-HSS moment connections under seismic loads. A welded connection incorporating plates that allowed an acceptable energy dissipation in the beam and the connection is performed. However, this connection was fully welded, requiring the use of field welding.

Another connection study by [21] performed experiments on connections to the column's weak axes, using a plate welded to the column flange. The results showed a higher moment resistance and increase of initial stiffness for this connection respect to connections welded directly to the column web. A research performed by [22] studied the behavior of welded connections between wide flange beam and box section columns through the addition of plates in the flanges and the web of the beam and field welding and [23] studied a new moment connection similar to welded unreinforced flange with box column through full-scale tests. The results showed that this connection to box column satisfied the requirements for special moment frames according to [6].

Hence, there have been no studies of a bolted moment connection between wide flange beam without reduction of flanges, to box section column without concrete fill considering two or more beams connected, bidirectional loading, and different axial load levels in the column.

This paper describes the research conducted to study the cyclic response of end-plate moment connections between wide flange beams and box columns. For this purpose, numerical models of end-plate moment connections, that were calibrated from previous tests [10], were generated for different joint configurations expanding to 3D joints, 2D joints and several levels of axial load. Previous research has focused on welded connections between wide flange beam and tubular column (hollow structural section or Box section), tubular beam to tubular column (welded for this type) and wide flange beam to wide flange column (welded and bolted for this type).

## 2. Description of Moment Connection

In this study, a typical end-plate moment connection was used as alternative for moment connections in Steel Moment Frames with wide flange beams, with the difference in the shape column (box section against wide flange). The numerical models were calibrated using the experimental

results reported in [10]. The end-plates were connected by high strength bolts to a similar plate connected through external diaphragms to the box column (Figure 1).

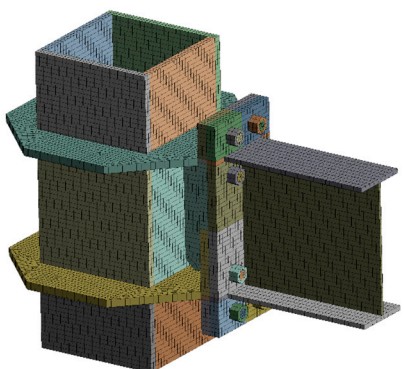

**Figure 1.** View of box section moment connection.

A combination of fillet and complete joint penetration welds was used between external diaphragms to the column and end-plates respectively. This configuration allowed optimizing the erection process in the field because the beams were bolted on site, avoiding field welding. Additionally, various configurations of beam-column connection were studied, where axial load and biaxial bending were considered. The beam end-plate can be sized following the recommendations for end-plate connections in [7]. According to [7], an analytical expression to calculate the capacity of the end-plate in the connection can be obtained using Yield Line Theory, through virtual work.

### 3. Finite Element Model

A numerical analysis using finite element method in ANSYS software [24], with different configurations of beam-column node was performed. In the analysis, axial load levels were considered simultaneously with the biaxial effect, as shown in Table 1 and Figure 2. The numerical models considered material nonlinearities, geometric nonlinearities, contact nonlinearities, and boundary conditions.

The beam and the column were designed for a prototype building located in Santiago, Chile, considering special moment frames as the main seismic resistant system with columns and beams of high ductility as established by [6]. The elements of the connection were designed as follows: the bolts based on the expected beam bending capacity. The bolts were the critical component in the connection design, since the end plates were designed to avoid prying and the welds of the beam and diaphragms to the end plates were assumed as CJP welds.

The horizontal diaphragms were designed from the maximum force due to tension capacity of the beam flange and the width-to-thickness ratio to prevent local buckling when the loads was reversed. The vertical diaphragm was designed to resist the shear transferred by beam capacity to the column (Figure 3a,b). Finally, the strong-column/weak-beam criteria was used incorporating a factor of 0.67 for the column capacity, taken from the equation (10.8-3) in [7] for the ConXtech® ConXL™ connection. This factor considers a reduction in column capacity due to the bidirectional bending on the column. No further details are given in [7] or [8]. A preliminary model in FEM was calibrated including dimensions and other specifications of specimens and boundary and loading conditions according to [10]. In the model, one beam to column joint configuration was analyzed. To reduce the computational cost, a BEAM188 element with two nodes was employed in beam and column outside of connection region. The elements of connection such as bolts, nuts, end-plate, vertical and horizontal stiffeners were modeled with SOLID186 element and nonlinearities of material, geometric and contact were considered. Finally, the hysteresis and moment-rotation curve were compared with experimental data of specimen #2, obtaining an acceptable agreement was

achieved between the finite element analysis results and experimental specimen #2, as shown in Figure 4.

### 3.1. General Characteristics of the Numerical Model

The models were analyzed with the following considerations: (1) the length of the column between inflection points of each story was 4.00 (m), (2) the welds were not included in the model considering that inelastic incursion was not expected in these elements and (3) the diameter of the holes was assumed equal to the diameter of the bolts. If the welds are incorporated into the model, a distribution of stress and deformation in welds could be obtained, avoiding concentration of stresses in critical areas. However, an elastic behavior was expected because the welds had a tensile strength greater than the tensile strength of base material used and also were designed for the capacity of the elements connected. Therefore, they were not modeled because the affect the response of the models is not affected and the computational cost is reduced. This last assumption considered the pretension in the bolts connecting the parts and was verified by [10].

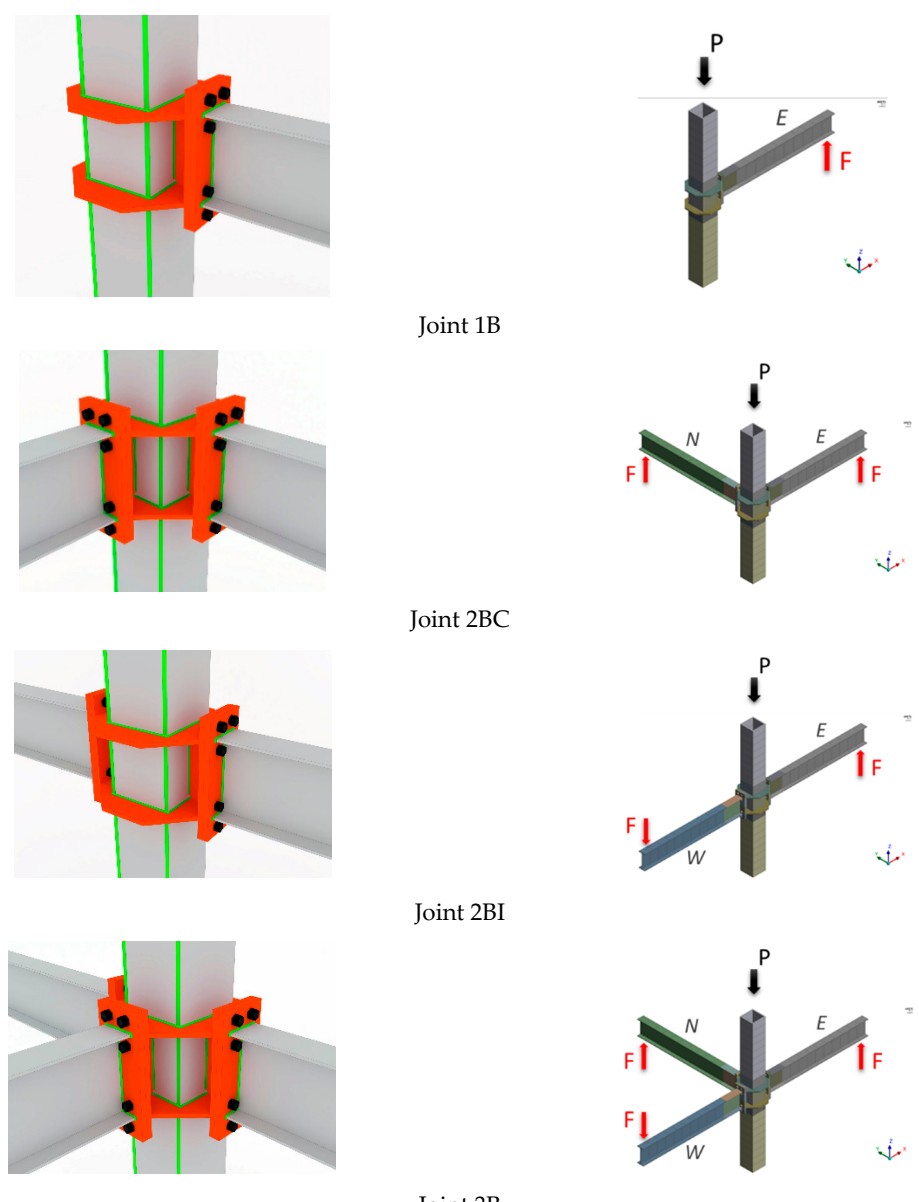

Joint 1B

Joint 2BC

Joint 2BI

Joint 3B

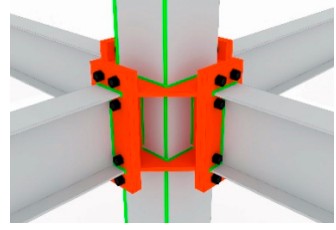
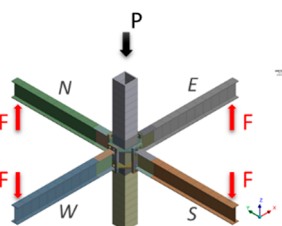

Joint 4B

**Figure 2.** Configurations of joint connections studied.

### 3.2. Element Type and Mesh

In the numerical study, hexahedral and tetrahedral 3D solid elements (SOLID 186) were used to model stiffeners, plates, bolts, beams, column, and nuts. The SOLID186 element has 20 nodes with three translational degrees of freedom per node, and can be used for materials with plasticity, hardening, and large deflections. The nut and the head of the bolt have similar diameter and thickness according to [25].

**Table 1.** Matrix of simulation in finite element models (FEMs).

| No. | Group | Joint | Axial Load (P/Py) |
|---|---|---|---|
| 1 | | 1B - 00 | 0 |
| 2 | 1 beam | 1B - 25 | 25% |
| 3 | (1B) | 1B - 50 | 50% |
| 4 | | 1B - 75 | 75% |
| 5 | | 2BC - 00 | 0 |
| 6 | 2 beams - corner | 2BC - 25 | 25% |
| 7 | (2BC) | 2BC - 50 | 50% |
| 8 | | 2BC - 75 | 75% |
| 9 | | 2BI - 00 | 0 |
| 10 | 2 beams - interior | 2BI - 25 | 25% |
| 11 | (2BI) | 2BI - 50 | 50% |
| 12 | | 2BI - 75 | 75% |
| 13 | | 3B - 00 | 0 |
| 14 | 3 beams | 3B - 25 | 25% |
| 15 | (3B) | 3B - 50 | 50% |
| 16 | | 3B - 75 | 75% |
| 17 | | 4B - 00 | 0 |
| 18 | 4 beams | 4B - 25 | 25% |
| 19 | (4B) | 4B - 50 | 50% |
| 20 | | 4B - 75 | 75% |

Note: $Py = FyAg$, where $Fy$ is yield stress and $Ag$ is the gross area of section

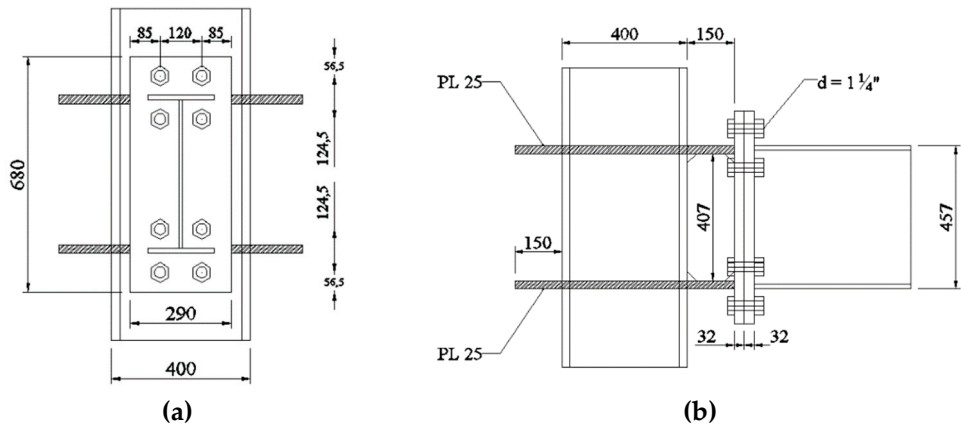

**(a)**          **(b)**

**Figure 3.** (**a**) Details of end-plate (mm) and (**b**) elevation view of moment connection (mm).

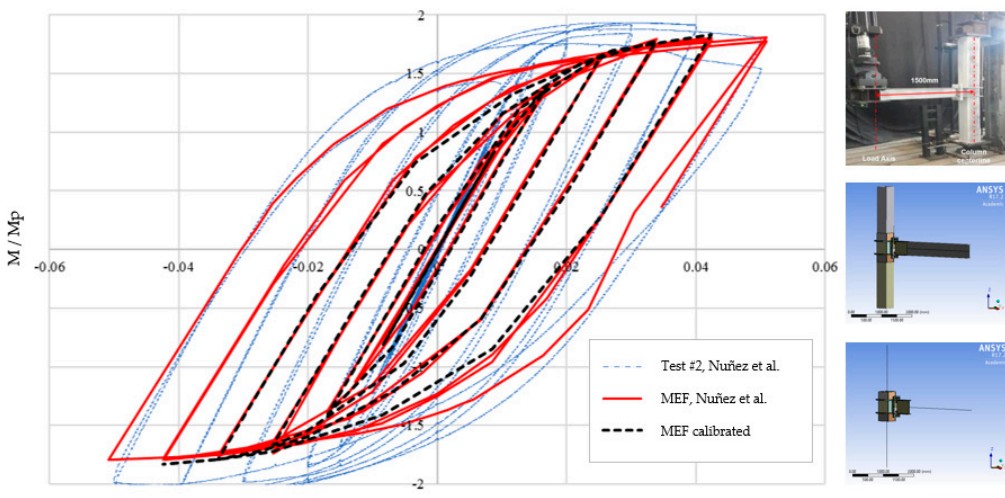

**Figure 4.** Model in FEM calibrated according to [10].

To achieve computational efficiency and convergence, a fine mesh was used in zones where large inelastic incursions were expected and a coarser mesh was used in the other zones. Additionally, a BEAM188 element with two nodes with six translational degrees of freedom per node was employed in the regions of beams and columns that were expected to remain elastic, to reduce the number of equations to solve, as shown in Figure 5. In summary, a 3D model with six Degrees of Freedom per node was used and each component was comprised of the following number of nodes: beam (11,594), column (45,136), outer diaphragms (17,958), vertical diaphragm (2003), end-plate (22,754), bolts and nuts (8234).

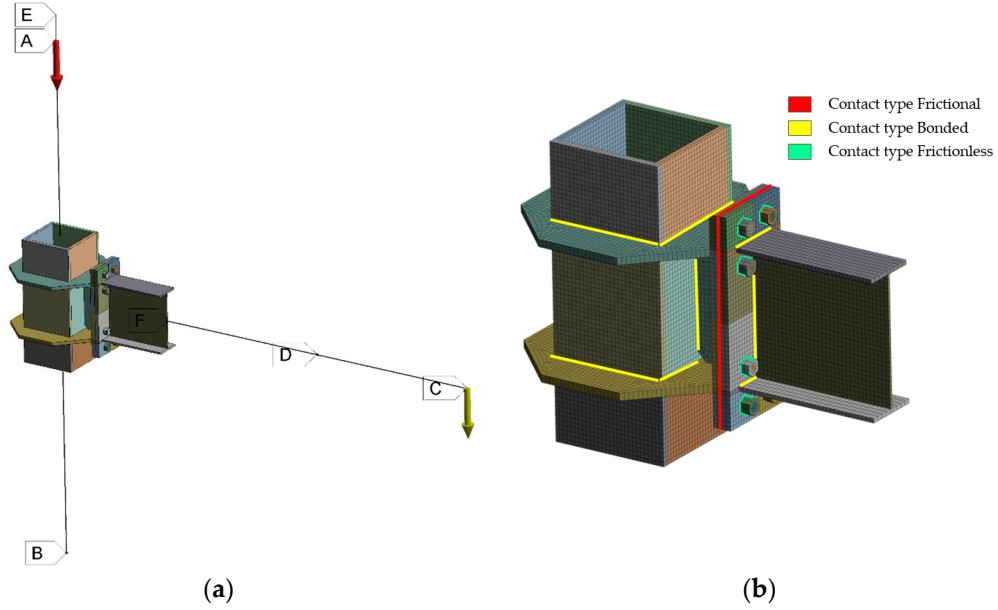

(**a**)  (**b**)

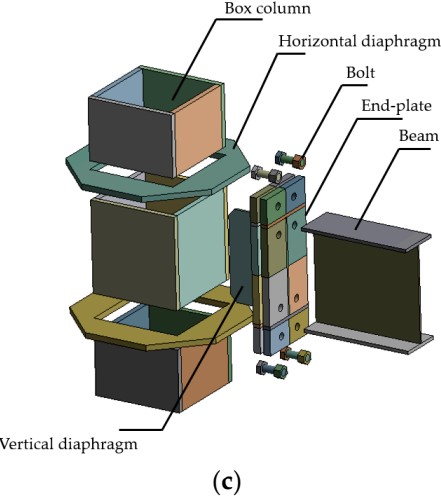

(**c**)

**Figure 5.** (**a**) Boundary conditions in numerical model, (**b**) contacts type in numerical model and (**c**) elements in moment connection.

### 3.3. Boundary Conditions, Contacts and Loading

As shown in Figure 5a, boundary conditions similar to those used in the tests were applied to the FEM models. The ends of the columns were articulated with restraints in three directions and the displacements applied at the end of the beam (s), according to the load protocol established in [6] and reproduced in Table 2. These conditions were applied using the "Remote Point Displacement" command. Additionally, a bolt pretension of 70% of the nominal tensile strength of the bolts, as specified in [6] was applied. The weld between elements was modeled using contact type "Bonded".

The FEM considers the interaction between the beam, column, bolts, bolt holes, vertical and horizontal diaphragms, nuts and end-plate. The friction coefficient between the end-plates was 0.3, a value found to be adequate according to [10]. Studies by [26] showed variations in the connection moment capacity of less than 2% for values within 0.1 and 1. A "Frictionless" contact was used between bolts and nuts, which allows separation between the connected parts and the tangential movement without considering the friction, according to research conducted by [27]. The load was introduced quasi-statically by means of several analysis steps in FEM. The bolt pretension was applied in the bolts before to load the model and vertical displacements are applied at the end of the beam according to [6]. The contacts type and elements used are shown in Figure 5b,c and summarized in Table 3. In the contacts employed, the Augmented Lagrange method was used to obtain numerical convergence in the contact region [25].

**Table 2.** Load protocol in FEM models.

| No. | No. of cycles | Drift angle ($\theta$) (rad) |
|-----|--------------|------------------------------|
| 1 | 6 | 0.00375 |
| 2 | 6 | 0.005 |
| 3 | 6 | 0.0075 |
| 4 | 4 | 0.01 |
| 5 | 2 | 0.015 |
| 6 | 2 | 0.02 |
| 7 | 2 | 0.03 |
| 8 | 2 | 0.04 |

Note: continue loading at increments of $\theta$ = 0.01 (rad), with two cycles of loading at each step

**Table 3.** Summary of contacts used in the numerical models.

| Elements Connection | Contact | Movement in Normal Direction | Movement in Tangential Direction |
|---|---|---|---|
| Column-Horizontal Stiffeners | Bonded | No separation | No slip |
| Column-Vertical Stiffeners | Bonded | No separation | No slip |
| Vertical Stiffeners-Horizontal Stiffeners | Bonded | No separation | No slip |
| End Plate-Horizontal Stiffeners | Bonded | No separation | No slip |
| End Plate-Vertical Stiffeners | Bonded | No separation | No slip |
| End Plate-End plate | Frictional | Separation allowed | Slip allowed |
| Beam-End plate, Bolt-Nut | Bonded | No separation | No slip |
| Bolt- End plate, Nut-End plate | Frictionless | Separation allowed | Slip allowed |

### 3.4. Material Modeling

The FEM includes two different types of steel: for beams, columns, vertical and horizontal stiffeners an ASTM-A36 material was employed and for bolts an ASTM-A490 material was used. In both cases the values are obtained from coupon tests (see Table 4). The stress-strain relationships with bilinear forms by means of kinematic hardening rule with von Mises yielding criterion was used to simulate metal plasticity (Figures 6 and 7).

**Table 4.** Material parameters of steel members from coupon tests.

| Element | Designation | $\sigma_y$ (MPa) | $\varepsilon_y$ | $\sigma_u$ (MPa) | $\varepsilon_u$ |
|---|---|---|---|---|---|
| Bolts | ASTM-A-490 | 1156 | 0.00586 | 1433 | 0.14 |
| Beam | ASTM-A-36 | 293 | 0.001465 | 445 | 0.24 |
| Column | ASTM-A-36 | 293 | 0.001465 | 445 | 0.24 |
| Horizontal diaphragms | ASTM-A-36 | 293 | 0.001465 | 445 | 0.24 |
| Vertical diaphragms | ASTM-A-36 | 293 | 0.001465 | 445 | 0.24 |
| End-plate | ASTM-A-36 | 293 | 0.001465 | 445 | 0.24 |

Notes: ($\sigma_y$), yield stress, ($\varepsilon_y$), yield strain, ($\sigma_u$), ultimate stress, ($\varepsilon_u$), ultimate strain

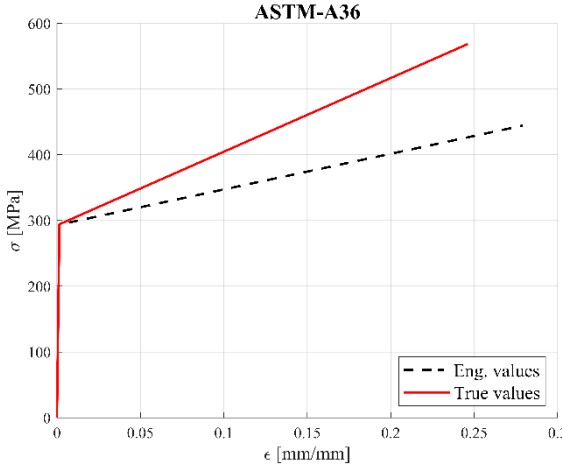

**Figure 6.** Simplified stress–strain relation of ASTM A36 material.

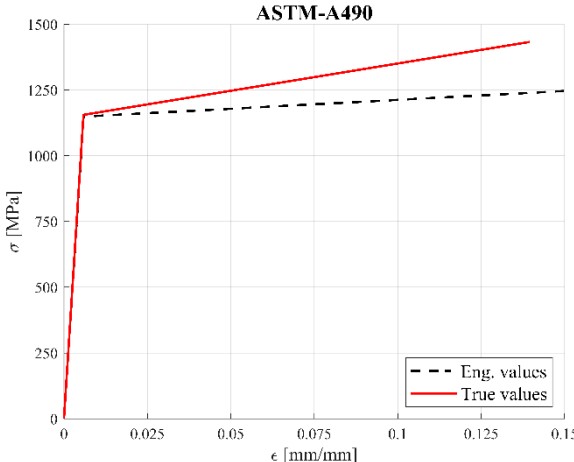

**Figure 7.** Simplified stress–strain relation of ASTM A490 material.

## 4. Analysis Results

In steel buildings with special moment frames the story drift angle of beam to column connections used in the seismic force resistant system should accommodate at least 4% drift ratio and a flexural resistance of the beam should be at least 0.80Mp. Likewise, the failure mechanism should be controlled by plastic hinges in the beams avoiding column failure prior to beam failure. In this research, the failure mechanism complies with the established in [6,7]. The results of numerical study are shown in next sections.

### 4.1. Seismic Performance for Different Joints

An equivalent load-displacement method for 2D and 3D joints is employed according to [9]. Based on the above equivalent column top hysteresis curves, the hysteresis behaviors of different joint configurations can be compared through stiffness, equivalent damping and energy dissipation [28]. This method allows the comparison of the seismic behavior between moment connections. The equivalent force in the top column ($V_c$) is the resultant force in horizontal direction and can be calculated using principles of structural mechanics. Equating the work performed by beam forces with work performed by equivalent force, an equivalent displacement (Δ) is obtained (Figure 8). The formulation of method in this research is shown as follows:

Initial System (Work performed by forces in all beams):

$$W_1 \ = \ \sum (F * \delta) \tag{1}$$

Equivalent System (Work performed by external force):

$$W_2 = \ V_C * \ \Delta \tag{2}$$

equating:

$$W_1 = \ W_2 \tag{3}$$

Joint 1B:

$$\alpha = 0° \tag{4}$$

$$Vc \ \ H = F \frac{L}{2} \ \ \Rightarrow \ \ Vc = \ \ \frac{1}{2} \frac{L}{H} \ \ F \tag{5}$$

$$Vc \ \ \Delta = F \ \ \delta_1 \ \ \Rightarrow \ \ \Delta = \ \ 2 \ \ \frac{H}{L} \delta_1 \tag{6}$$

$$\theta = \ \ ^{\Delta}\!/_H \tag{7}$$

Joint 2BC:

$$\alpha = 45° \tag{8}$$

$$Vc_X = Vc_Y = \frac{1}{2}\frac{L}{H}\ F \ \Rightarrow \ Vc = \frac{\sqrt{2}}{2}\frac{L}{H}\ F \tag{9}$$

$$Vc\ \Delta = F\ \delta_1 + \ F\ \delta_2 \ \Rightarrow \ \Delta = \frac{2}{\sqrt{2}}\frac{H}{L}(\delta_1 + \ \delta_2) \tag{10}$$

$$\theta = \Delta/_H \tag{11}$$

Joint 2BI:

$$\alpha = 0° \tag{12}$$

$$Vc_1 = Vc_2 = \frac{1}{2}\frac{L}{H}\ F \ \Rightarrow \ Vc = \frac{L}{H}\ F \tag{13}$$

$$Vc\ \Delta = F\ \delta_1 - \ F\ \delta_2 \ \Rightarrow \ \Delta = \frac{H}{L}(\delta_1 - \ \delta_2) \tag{14}$$

$$\theta = \Delta/_H \tag{15}$$

Joint 3B:

$$\alpha = 26.6° \tag{16}$$

$$Vc_x = \frac{L}{H}\ F \ ; \ Vc_y = \frac{1}{2}\frac{L}{H}\ F \ \Rightarrow \ Vc = \sqrt{1.25}\frac{L}{H}\ F \tag{17}$$

$$Vc\ \Delta = F\ \delta_1 - F\ \delta_2 + \ F\ \delta_3 \ \Rightarrow \ \Delta = \frac{1}{\sqrt{1.25}}\ \frac{H}{L}(\delta_1 - \delta_2 + \delta_3) \tag{18}$$

$$\theta = \Delta/_H \tag{19}$$

Joint 4B:

$$\alpha = 45° \tag{20}$$

$$Vc_x = \ Vc_y = \frac{L}{H}\ F \ \Rightarrow \ Vc = \sqrt{2}\frac{L}{H}\ F \tag{21}$$

$$Vc\ \Delta = F\ \delta_1 - F\ \delta_2 + \ F\ \delta_3 - \ F\ \delta_4 \Rightarrow \Delta = \frac{1}{\sqrt{2}}\ \frac{H}{L}(\delta_1 - \delta_2 + \delta_3 - \delta_4) \tag{22}$$

$$\theta = \Delta/_H \tag{23}$$

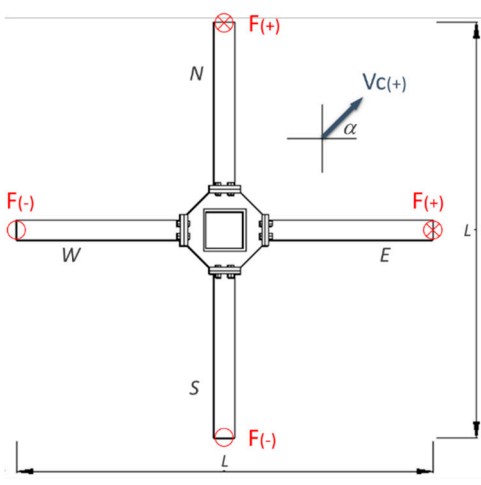

**Figure 8.** Equivalent load-displacement method.

## 4.2. Hysteretic Behavior

The results obtained shown a similar hysteretic curve for different beams in each direction for the same shape. Due to the large number of results (response of East, West, North and South beams) were obtained, uniquely the results for the East beam were reported. As shown in Figure 9, the model 1B, model 2BC, model 2BI, model 3B and model 4B reached a drift angle and flexural resistance of the beam greater than 4% drift ratio and 0.80 Mp, where Mp = 324 (kNm), respectively, is obtained. The stiffness and the resistance of connection was maintained.

A slight pinching at an axial load of 75% the column capacity is observed for 1B, 2BC and 2BI models. In models 3B and 4B the pinching greater than other models even where complies the requirements according to [9]. When an increase in the column axial load occurred, plastic strains in the column walls appeared, which explain the loss of resistance in the hysteretic behavior at an axial load of 75% the column capacity.

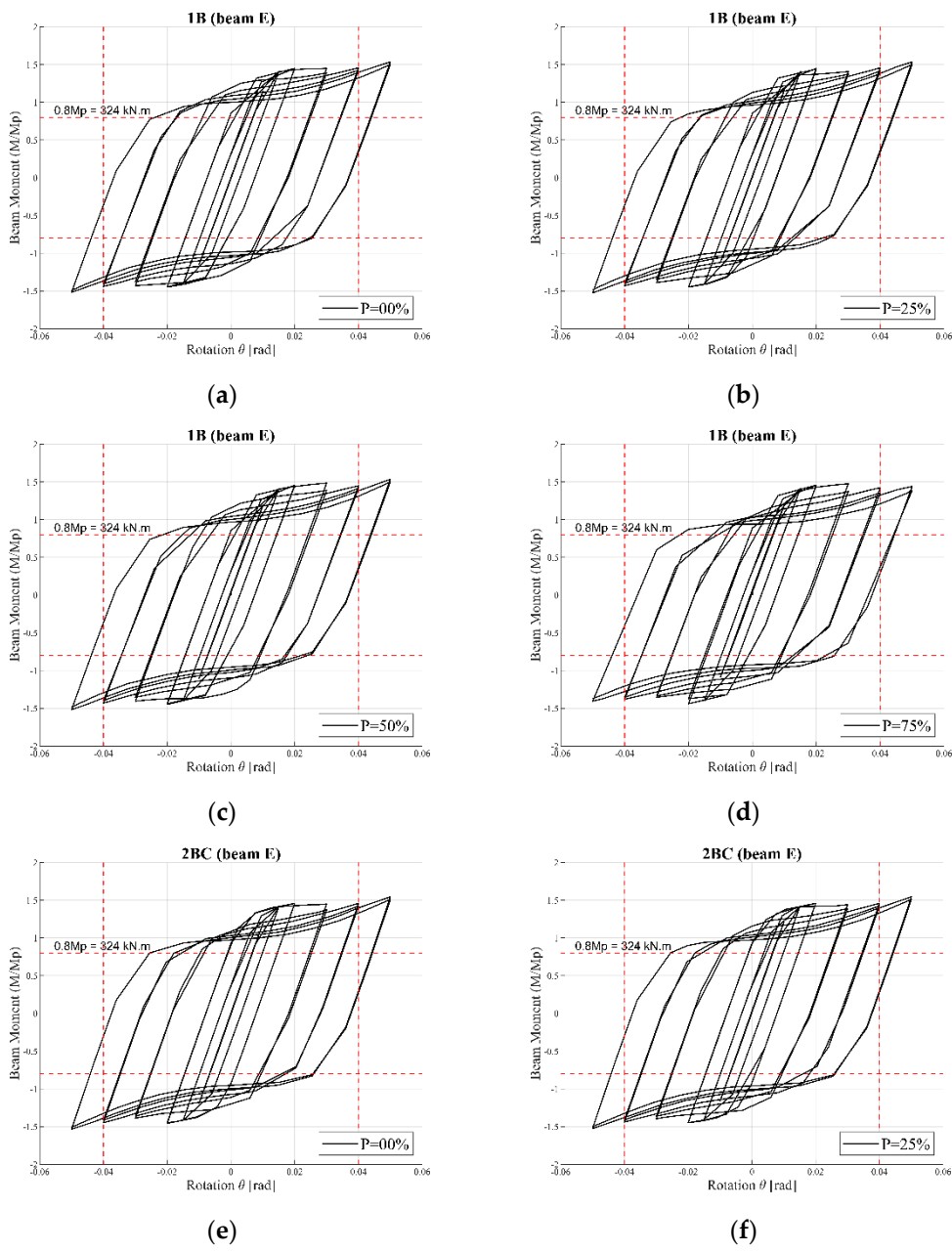

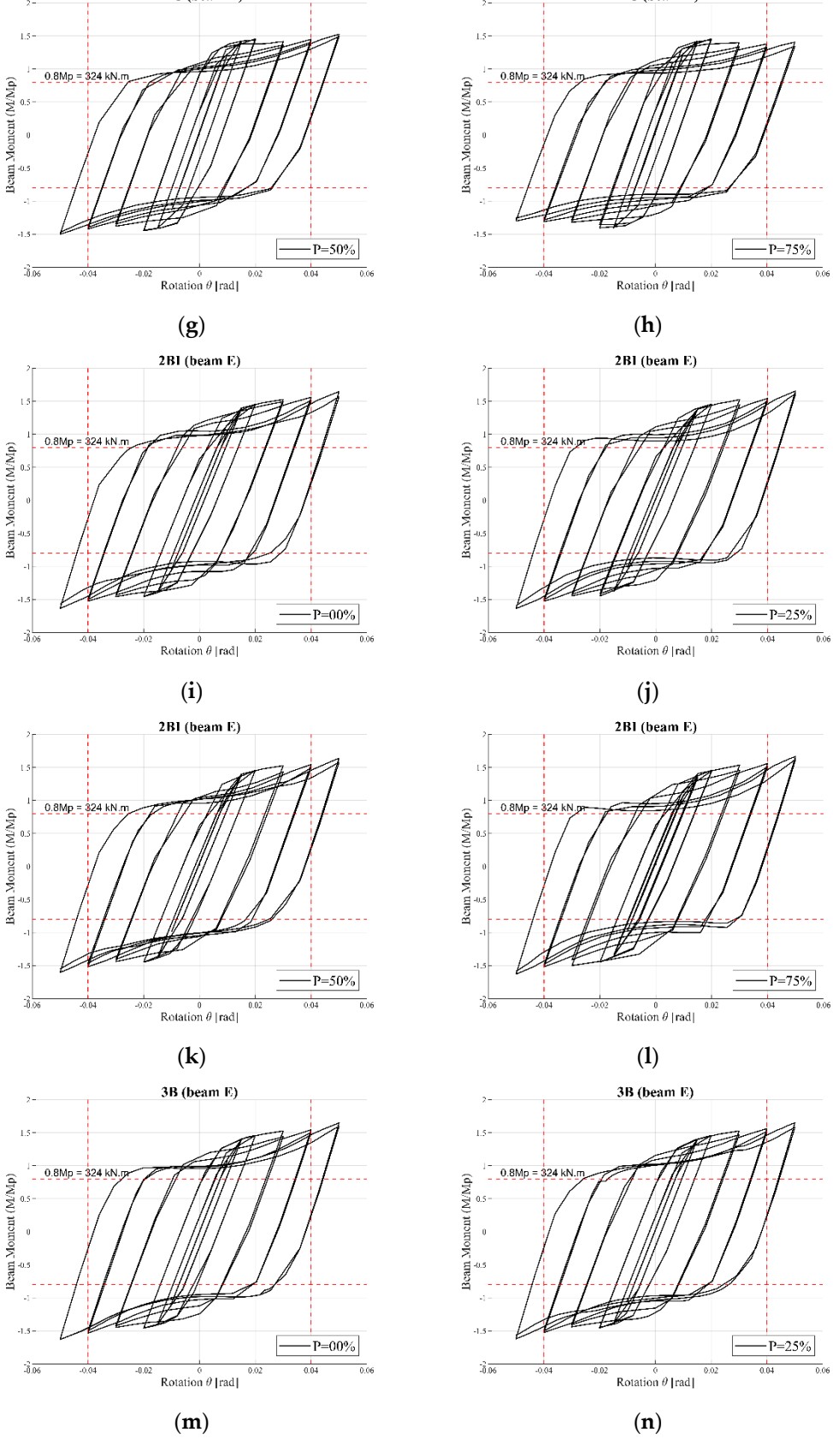

(**g**)　　　　　　　　　　　　　　　　　　　　　　　　(**h**)

(**i**)　　　　　　　　　　　　　　　　　　　　　　　　(**j**)

(**k**)　　　　　　　　　　　　　　　　　　　　　　　　(**l**)

(**m**)　　　　　　　　　　　　　　　　　　　　　　　　(**n**)

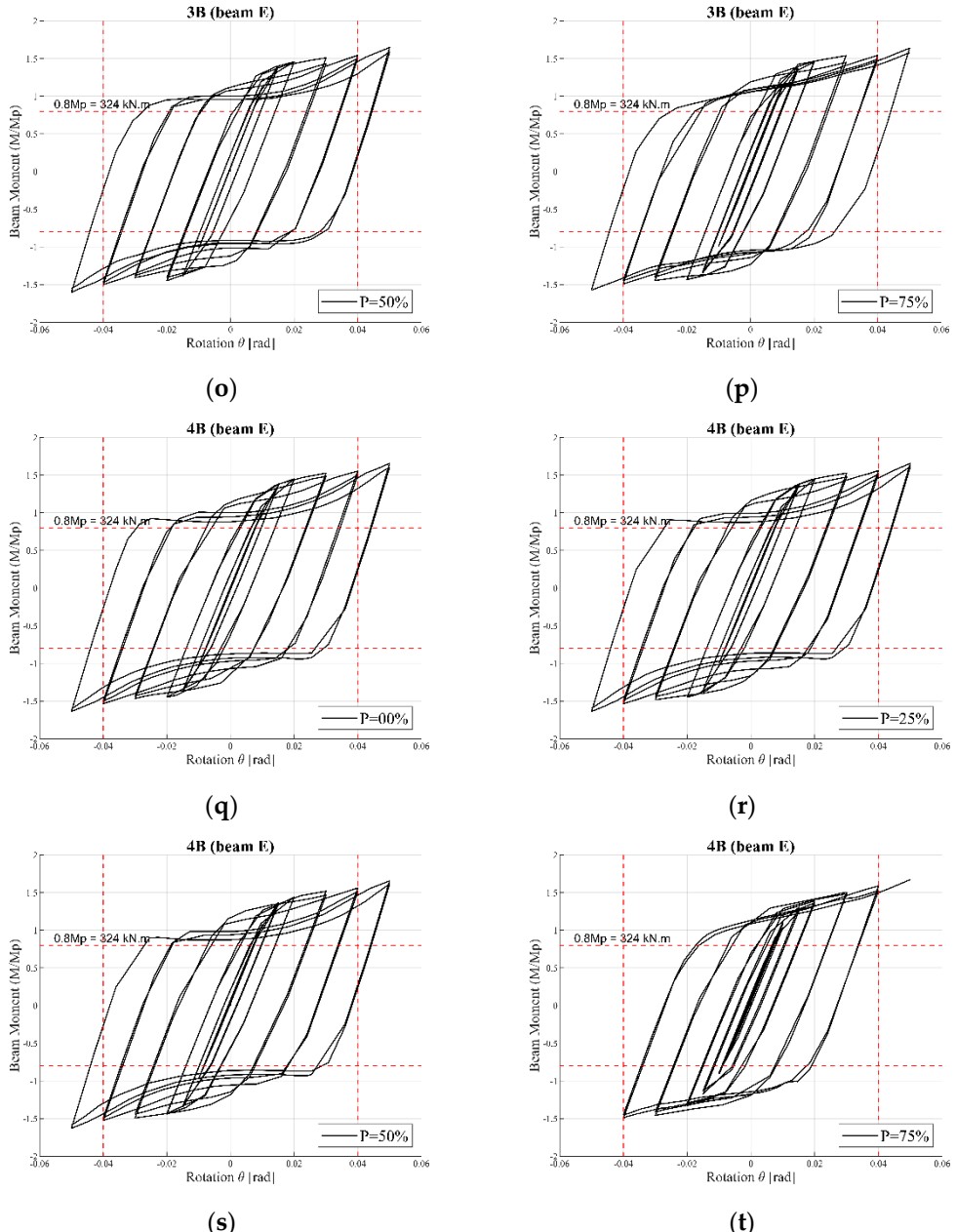

**Figure 9.** Summary of normalized moment-rotation at each east beam under different axial load levels in the column.

### 4.3. Failure Mechanism

The desirable failure mechanism in joints under seismic loads is flexural yielding of beams prior to column failure, as the strong-column/weak-beam requirement according to seismic provisions. A simultaneously failure could be obtained when beams in orthogonal direction were connected; both failure mechanisms were verified. The von Mises equivalent stress distribution and final deformation of each joint at the maximum load point are shown in Figure 10. All joints configuration developed the plastic hinge at beam as a failure mechanism; a plastic hinge at column as failure mechanism was achieved in 2BI, 3B and 4B cases where axial load was 0.75 Py. In the rest of the cases, the equivalent stresses were higher than the yield stress of the material but less than the ultimate stress.

(**a**)

(**b**)

(**c**)

(**d**)

(**e**)

(**f**)

(**g**)

(**h**)

(**i**)　　　　　　　　　　　　　　　　　　　　(**j**)

(**k**)　　　　　　　　　　　　　　　　　　　　(**l**)

(**m**)　　　　　　　　　　　　　　　　　　　　(**n**)

(**o**)　　　　　　　　　　　　　　　　　　　　(**p**)

(**q**)  (**r**)

(**s**)  (**t**)

**Figure 10.** Summary of von Mises stress distribution by the imposed maximum displacement in joints.

Plastic strains are reached exclusively in the beam for different axial loads except in the model 2BI with an axial load of 75% the column capacity, where plastic strains are observed in column. The model 3B and 4B has failure mechanism combined for an axial load of 75% the column capacity with incursion in plastic range of column as shown in Figure 11 (the blue region indicates that the strain is elastic range).

(**a**)  (**b**)

(**c**)

(**d**)

(**e**)

(**f**)

(**g**)

(**h**)

(**i**)

(**j**)

(**k**) (**l**)

(**m**) (**n**)

(**o**) (**p**)

(**q**) (**r**)

(**s**)                                                                (**t**)

**Figure 11.** Summary of plastic strains by the imposed maximum displacement in joints.

However, it is unlikely that there was an axial load of 75% the column capacity on columns that belonged to special moment frames that comply with the drift according to the standards. Therefore, column and connection elements showed stresses in elastic range, which is acceptable according to [9].

According to the load-displacement method, the hysteresis curve of five different joint configurations can be evaluated through stiffness, damping and energy dissipation parameters. As shown in Figure 12, a similar behavior in the five joints was obtained. Joint 2BC could sustain 25% more load with respect to Joint 1B, Joint 2BC could sustain 60% more load with respect to Joint 2BI and Joint 4B 25% more load with respect to Joint 2BI and Joint 3B. The rotation in Joint 1B and Joint 2BI reached 0.045 [rad] drift, while Joint 2BC, Joint 3B and Joint 4B reached 0.06 [rad] drift, demonstrating that rotation levels in 3D joints were greater than 2D joints. The behavior in all joints showed a pinching for an axial load of 75% pf the column capacity in column, which evidences a combined failure mechanism composed by simultaneous inelastic behavior in beams and column unlike other axial load levels.

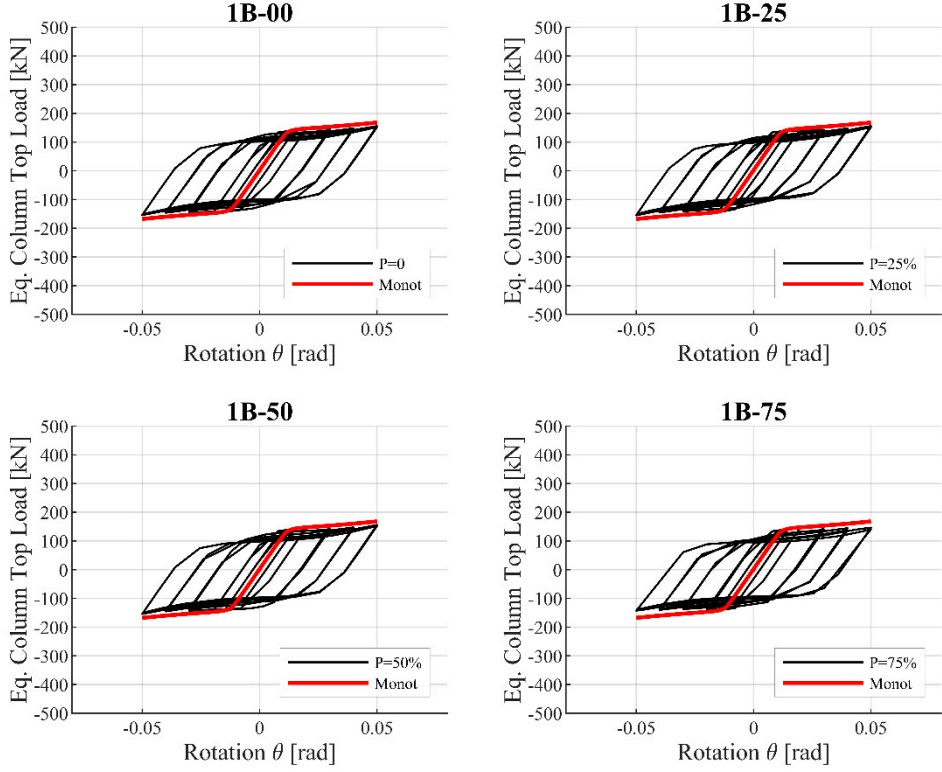

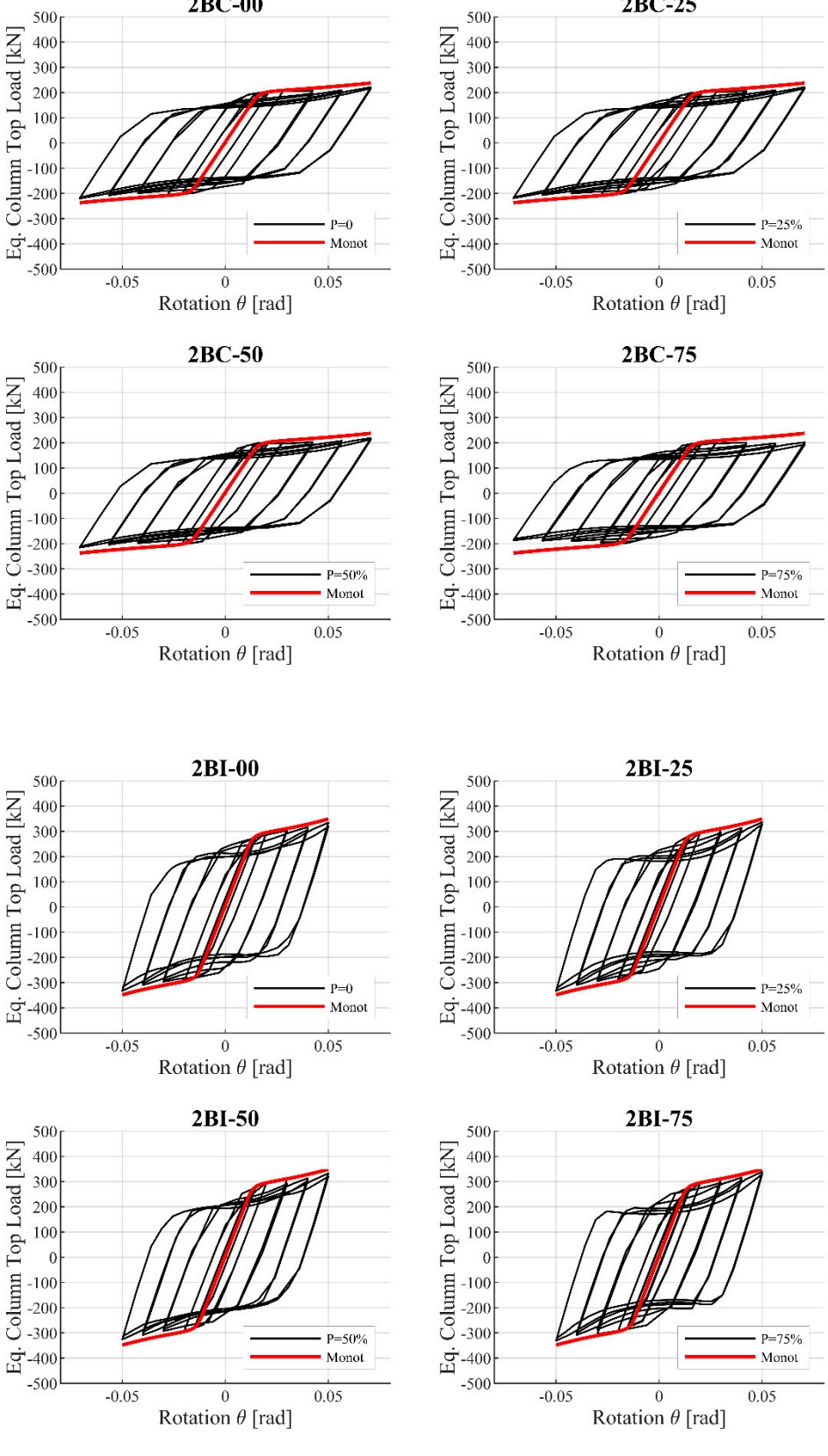

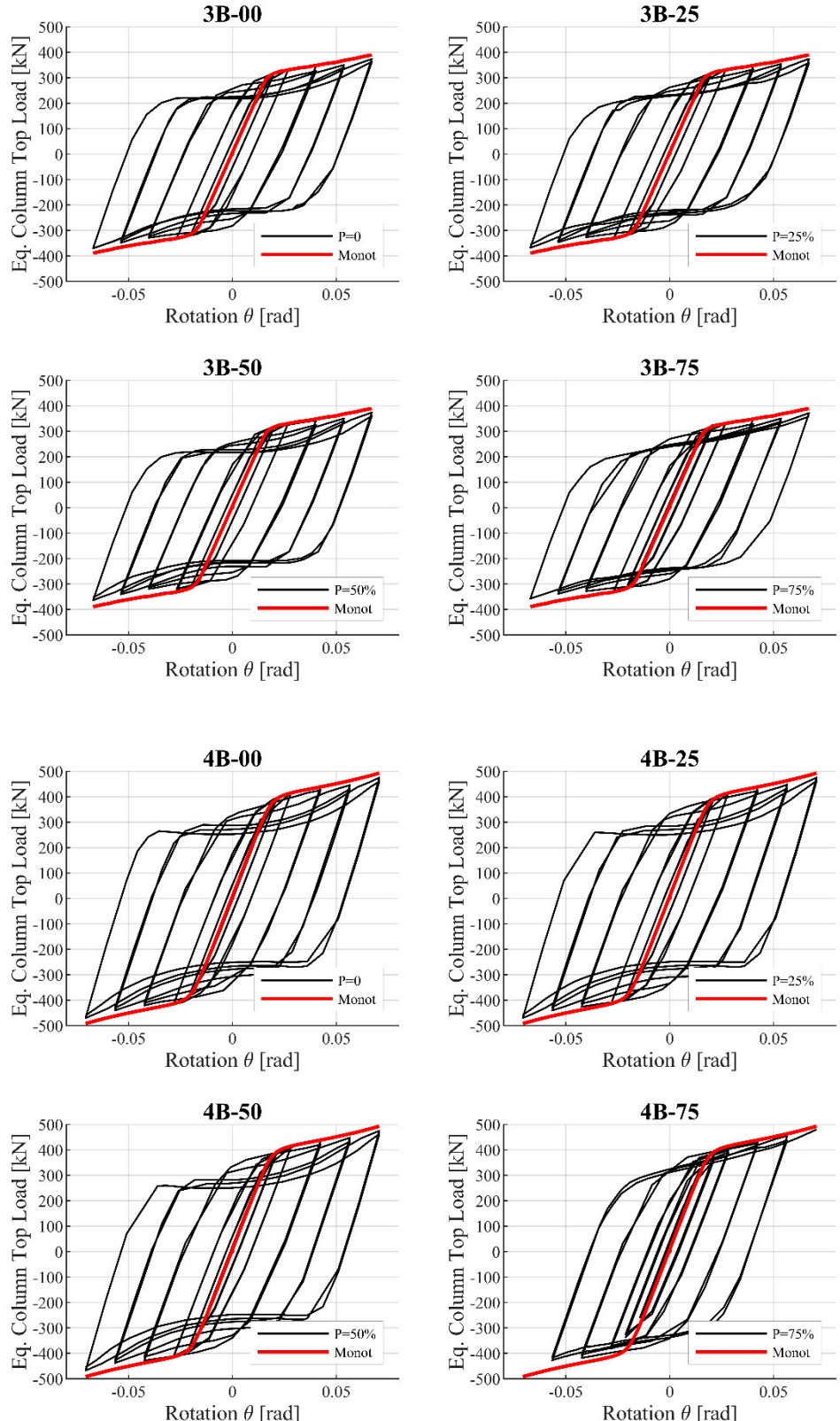

**Figure 12.** Summary of equivalent load vs according to rotation according to equivalent load-displacement method.

In Figure 13, the normalized tangent stiffness (slope for each loop in load or reload segment/slope for elastic loop) in joints is reported. Values near to 1 were observed for all joints

independently of the axial load level, showing that joint stiffness degradation in this moment connection was acceptable. Likewise, in the Figure 14 the normalized secant stiffness (slope of the line that joins a point of maximum load with the origin/slope for elastic loop) is reported. A 40% of stiffness was sustained for 0.04 [rad] drift angle in all joint configurations.

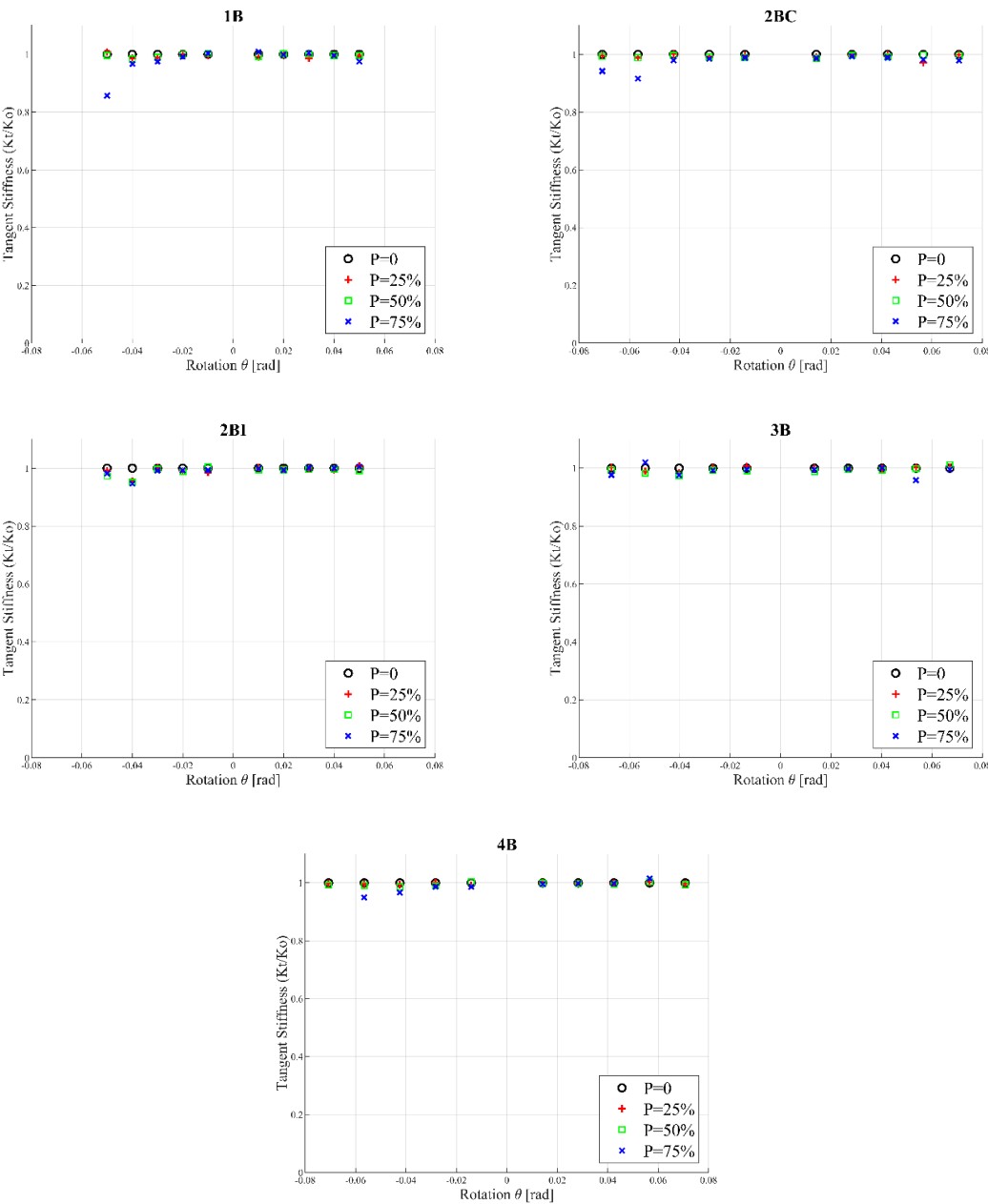

**Figure 13.** Summary of tangent stiffness vs. rotation according to equivalent load-displacement method.

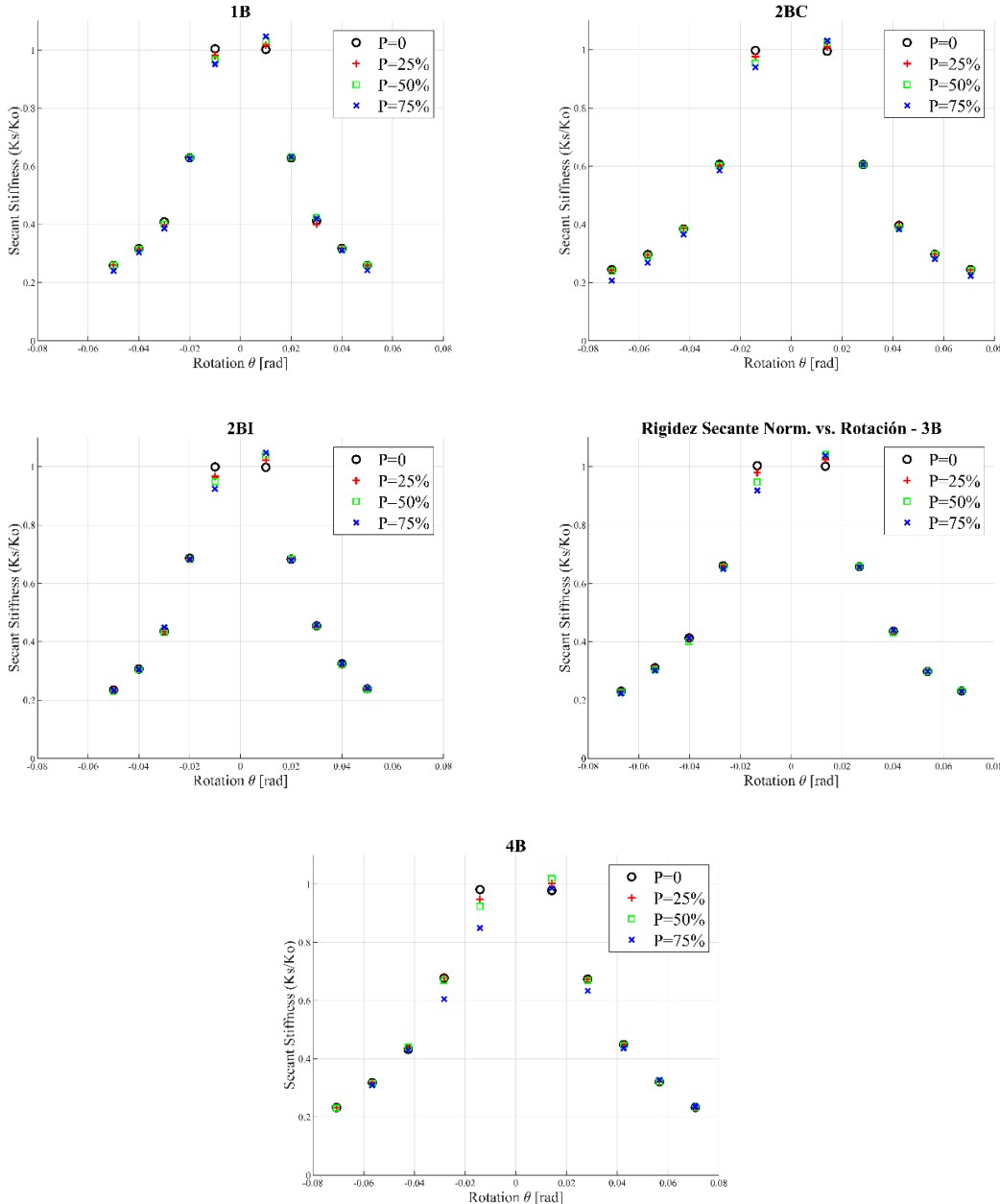

**Figure 14.** Summary of secant stiffness vs. rotation according to equivalent load-displacement method.

As shown in Figure 15, the dissipated energy of joints 3D was greater than joints 2D, due to 3D joints having a greater number of beams connected. Conversely, if the dissipated energy was normalized by number of beams connected, the joints 2D showed a higher dissipated energy. In the Figure 16, the equivalent damping for 0.01 [rad] drift was approximately 3% in all joints. These values are used in design of steel buildings generally. Additionally, for 0.04 (rad) drift a 33% equivalent damping was reported in joints 2BI, 3B and 4B. For joints 1B and 2BC a 35% of equivalent damping was obtained. In both cases, these results were evidence of damage. In Table 5, a summary of equivalent load-displacement curves results is reported.

**Table 5.** Summary of equivalent load-displacement curves results.

| Joint | Maximum Load | Maximum Rotation | Initial Stiffness | Dissipated Energy (kJ) | Maximum Equivalent Damping at 4% Drift Ratio (%) |
|---|---|---|---|---|---|

|         | (kN)   | (rad) | (kN/mm) |         |      |
|---------|--------|-------|---------|---------|------|
| 1B - 00 | 155.75 | 0.050 | 2.90    | 328.71  | 34.5 |
| 1B - 25 | 155.75 | 0.050 | 2.89    | 318.53  | 32.8 |
| 1B - 50 | 155.75 | 0.050 | 2.91    | 322.87  | 33.4 |
| 1B - 75 | 149.63 | 0.050 | 2.91    | 314.04  | 33.5 |
| 2BC - 00 | 221.50 | 0.071 | 3.02   | 666.78  | 32.8 |
| 2BC - 25 | 221.50 | 0.071 | 3.01   | 671.03  | 33.5 |
| 2BC - 50 | 219.03 | 0.071 | 2.99   | 658.58  | 32.8 |
| 2BC - 75 | 209.13 | 0.071 | 2.99   | 653.18  | 34.5 |
| 2BI - 00 | 334.25 | 0.050 | 5.25   | 662.04  | 32.2 |
| 2BI - 25 | 336.00 | 0.050 | 5.18   | 613.86  | 31   |
| 2BI - 50 | 332.50 | 0.050 | 5.27   | 623.54  | 30.5 |
| 2BI - 75 | 337.75 | 0.050 | 5.21   | 609.25  | 30.5 |
| 3B - 00 | 373.70 | 0.067 | 4.54    | 976.54  | 31   |
| 3B - 25 | 373.70 | 0.067 | 4.57    | 982.28  | 31.3 |
| 3B - 50 | 373.70 | 0.067 | 4.50    | 957.35  | 31.5 |
| 3B - 75 | 371.75 | 0.067 | 4.52    | 900.20  | 31.3 |
| 4B - 00 | 475.18 | 0.071 | 5.30    | 1251.40 | 30   |
| 4B - 25 | 475.18 | 0.071 | 5.26    | 1250.20 | 29.8 |
| 4B - 50 | 475.18 | 0.071 | 5.32    | 1247.00 | 29.5 |
| 4B - 75 | 480.13 | 0.071 | 5.23    | 947.20  | 30.5 |

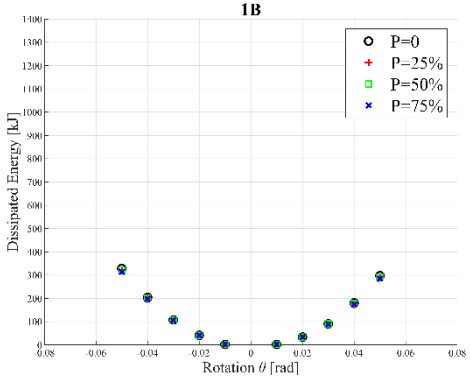

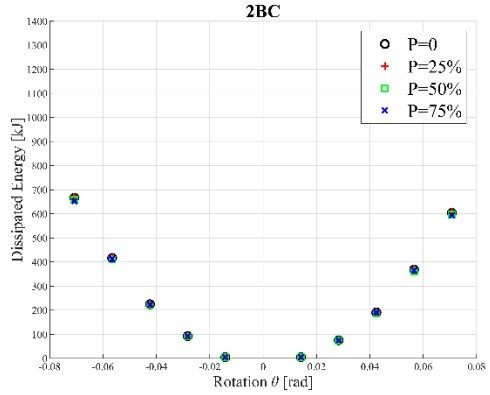

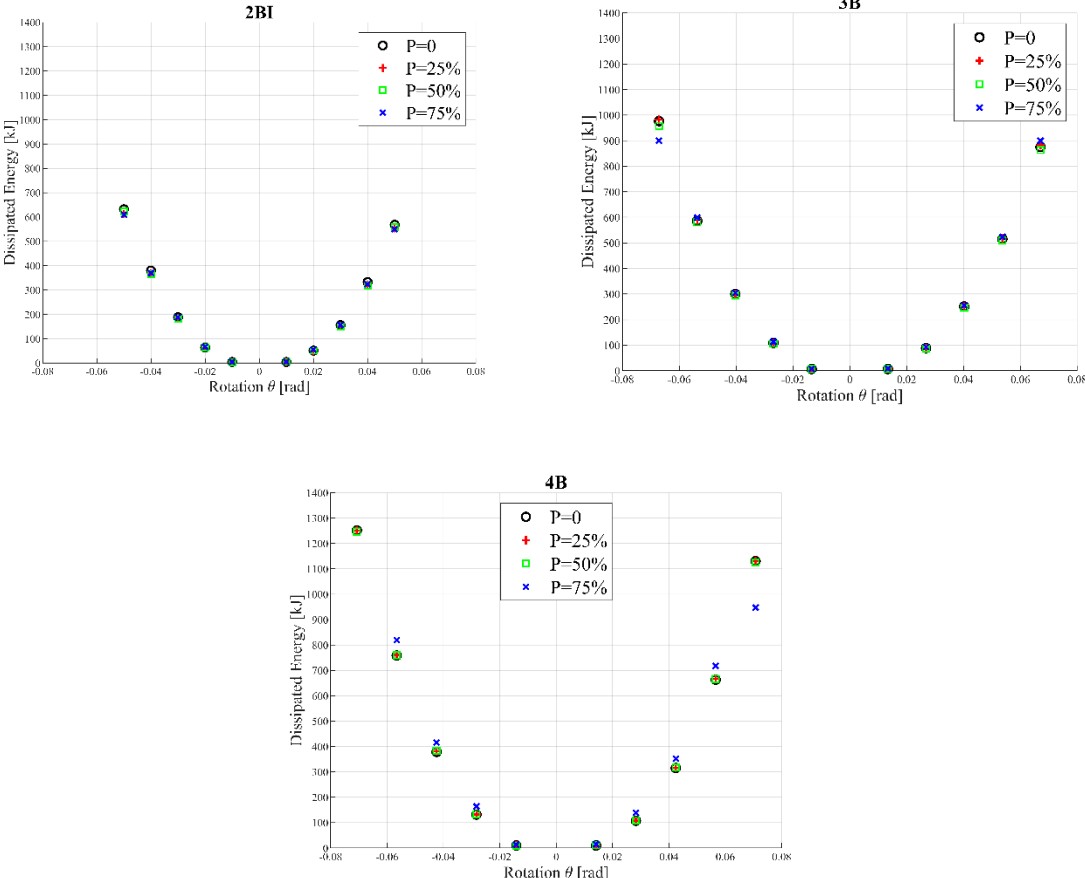

**Figure 15.** Summary of dissipated energy vs. rotation according to equivalent load-displacement method.

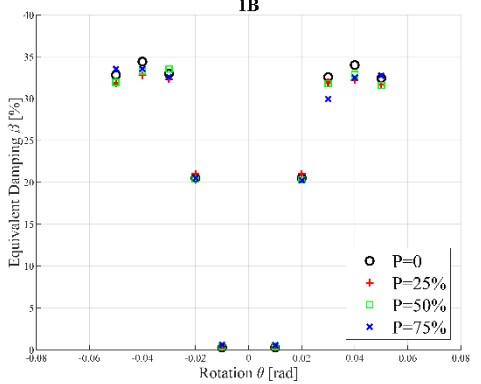
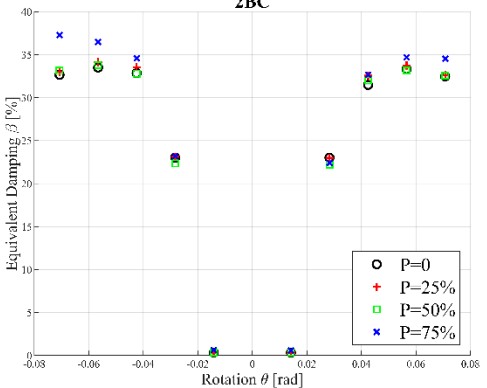

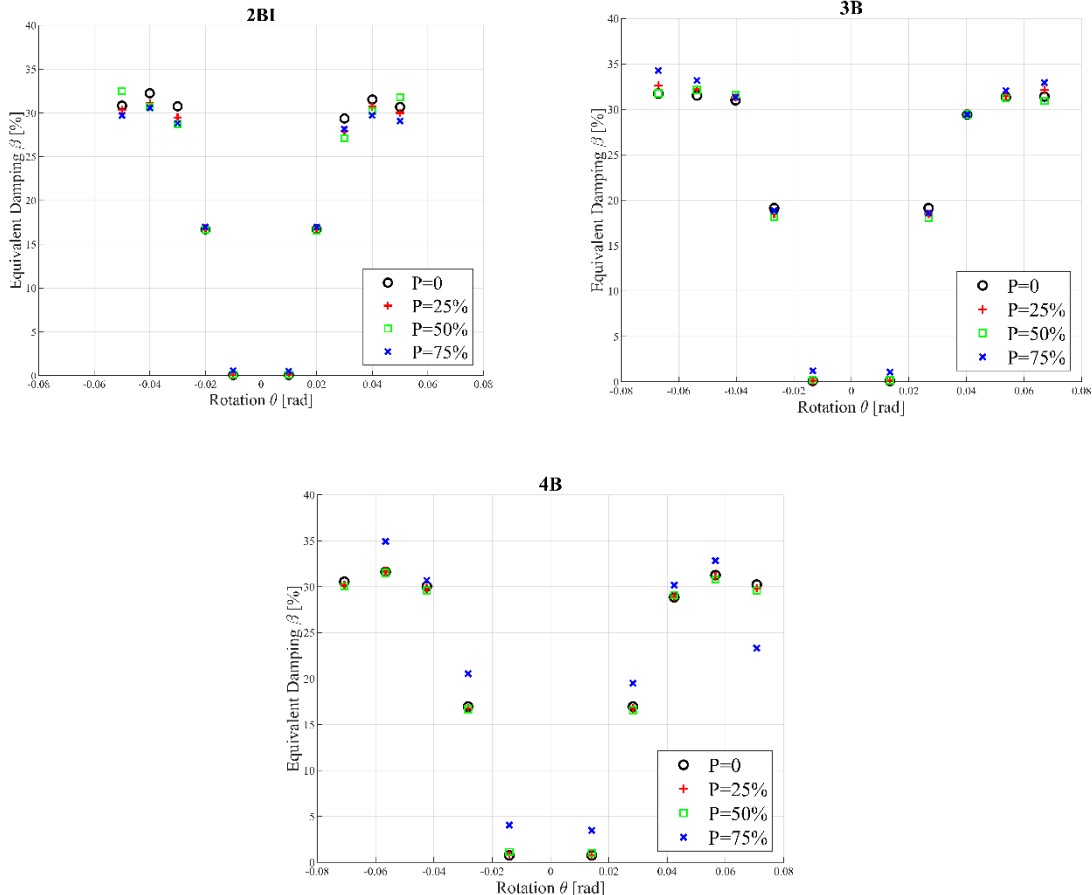

**Figure 16.** Summary of equivalent damping vs. rotation according to equivalent load-displacement method.

## 5. Conclusions

In this research, an extensive numerical study of seismic behavior of box-column connected with end-plate moment connection was developed considering 3D joints, which are more representative to real structures than 2D joints. This configuration of moment connection allows avoiding field welds through the incorporation of bolts, end-plate and external diaphragms and providing bidirectional resistance in orthogonal directions due to use of box-columns. The numerical study in FEM based in ANSYS software capture the hysteretic behavior of five types of joints: 1B (exterior), 2BI (interior), 2BC (exterior), 3B (exterior) and 4B (interior) using a total of 20 configurations with different levels of axial load. The main conclusions are as follows:

(1) The moment connection studied complies with the design philosophy, failure mechanisms and behavior established by Seismic Provisions. For all joint configurations a drift angle in excess of 0.04 (rad) and a flexural resistance greater than 0.8 Mp for all axial load levels is achieved.

(2) The failure is concentrated in the beams of all joint configurations except in joints 3D and 2D for an axial load of 75% the column capacity, where a combined failure mechanism (plastic deformations in beam and column) is achieved, showing a good correlation with the strong column/weak beam criteria developed for the Contech® ConXL™ connection.

(3) The elements outer diaphragm and vertical diaphragm, end-plate and bolts remain in elastic range while the beam reach the maximum inelastic incursion until 5% interstory drift. Additionally, the several components of connection around to panel zone avoid their distortion.

(4) The equivalent damping and dissipated energy is similar for the five joint types, being greater in the 3D joints than the 2D joints. The global deformation in 3D joints is greater than 2D joints even with the same number of beams. Therefore, the response of joints may be underestimated if 3D effects are not considered.

(5) The axial load level and bidirectional effect simultaneously affect the performance of joints as can be observed in the loss of energy dissipation. However, hysteretic loops without significant pinching are observed for all joint configurations.

## Nomenclature

| $\alpha$ | Angle from $V_c$ |
|---|---|
| $\Delta$ | Equivalent column top displacement |
| $F$ | Vertical load at the beam end |
| $H$ | Distance between zero moment points |
| $L$ | Distance between the loading points |
| $M_p$ | Plastic moment of beam |
| $V_c$ | Equivalent force in the top column |
| $V_{cx}$ | Equivalent force in the top column in X direction |
| $V_{cx}$ | Equivalent force in the top column in Y direction |
| $W_1$ | Work performed by loads at beam end |
| $W_2$ | Work performed by Vc force |
| $\delta_i$ | (i = 1,2,3,4) vertical displacement at the beam for east, west, north and south |
| $\varepsilon_u$ | Ultimate deformation |
| $\varepsilon_y$ | Yielding deformation |
| $\theta$ | Rotation angle due to moment of beam |
| $\sigma_u$ | Ultimate stress |
| $\sigma_y$ | Yielding stress |

**Author Contributions:** Conceptualization, (E.N.) and (R.H.); methodology, (E.N.); software, (M.G.); validation, (E.N.) and (R.H.); formal analysis, (M.G.); investigation, (E.N.), (R.H.) and (M.G.); writing—original draft preparation, (E.N.); writing—review and editing, (R.H.); visualization, (R.H.); supervision, (R.H.).

**Funding:** This research received no external funding.

**Conflicts of Interest:** The authors declare no conflict of interest.

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
