# Peer review of "Numerical Study on Cyclic Response of End-Plate Biaxial Moment Connection in Box Columns"

_metals, doi:10.3390/met10040523_

Round 1
Reviewer 1 Report
The present artcile is within Journal's aims. I found it extremely interesting from both theoretical and practical point of view. The paper can be published after some minor modifications and amendments in order to clarify some aspects and help the casual readers.
1) In abstract the authors say 'only one proprietary' and the 'rest of connections'. You have to name the proprietary connections or say the rest connections that are not proprietary.
2) In Introduction (lines 40-43), tubular sections possess higher axial capacity in comparison to wide flange sections. Please note that both types of sections may have similar moment of inertia (major bending axis), but it is the axial capacity that renders tubular sections a better option than wide flanges. On the other hand, beam-to-column connections are easier (in terms of fabrication) when wide flange columns are used instead of tubular columns. The statement regarding lateral bracing is correct, but in usual practical cases, lateral buckling is more crucial for beams and not for columns. Deterioration is affected by the duration of seismic motion and thus, I do not find any reason to relate torsional stiffness with deterioration (line 43). Thus, please modify these lines.
3) Line 63: mention the axial load level.
4) Line 67: 5% drift is not acceptable by any practicing engineer. I understand what the authors want to convey but please have in mind that seismic design codes even for steel MRFs do not accept such drifts which because in all likelihood surpass the collapse prevention performance limits. Storey stability and serviceability drift checks will also lead to lower drift values because of the final sections chosen. That is, the fact that experiments on connections reveal a 5% drift capacity does not mean that the whole structure can accommodate such drift levels without exhibiting serious damage. Please comment this.
4) Introduction has to be ordered. Your contribution should be mentioned right after the complete literature review.
5) In line 151 you mention geometric nonlinearities and then large deflections. I cannot understand the difference. Are we talking about initial imperfections or cambers (geometric nonlinearities) and large strains? Please clarify.
6) In line 157 you have to mention the element (end plate, weld etc.) which is the critical one (in terms of design resistance) of the connection.
7) In line 160: you mention tension force and then local buckling. How can local buckling be induced by tension?
8) In line 162: is there any physical explanation for this 0.67 value?
9) In lines 167-168: I understand why you did not model the welds. What is to be expected if welds are modelled?
10) In line 177, choose another verb to replace 'admit'
11) Py is Table 1 is the design axial strength (after buckling checks) and not the critical axial load. Please say that in text otherwise some readers will get confused.
12) Lines 207-210. Have you tested different values for the friction coefficient? If so, is there any difference if we vary this coefficient between 0.2-0.45?
13) The connection elements mentioned in Table 3 have to be shown in Figure 4 or in Figure 1.
14) Please use subscripts for y, u to ε, σ in Table 4. Units of ε can be omitted.
15) Line 236: Use KNm also in parentheses. Same in line 289.
16) Line 243: equivalent damping comes from the area of the hysteresis loop. The integral of this area gives the dissipated energy. Put a reference there.
17) Line 246: say mechanics and not mechanical
18) I believe that lines 258-284 can be written in a more compact form. Use α as a parameter and express everything in terms of α. Then make a comment for the usual cases of 0 and 45 degrees.
19) Provide equivalent damping in Table 5 next to dissipated energy. Thus, for every joint type, an equivalent damping estimate will be available. This is important because dissipated energy is not part of any design code, but equivalent damping (even in global sense) is.
20) Mention in Introduction and conclusions that 3D joints are closer to real praxis than 2D ones.
Reviewer 2 Report
The authors provided detailed Finite Element model analyses of a steel beam-to-column joint subjected to multiple combinations of loads. The presented joints fulfill the requisites provided by the ANSI seismic provisions.
The literature review is well organized and consistent. Details on the FE model are given. The main critic to this work is given to the model calibration, which was mentioned several times by the authors along the manuscript but it was not reported. Authors refers to a previous published work and, in the reviewer's opinion, a section describing the model calibration/validation should be added to give real "consistency" to the presented work.
Results are generally well presented and not questionable. Unfortunately some of the graphs (e.g., Fig. 9 to Fig.12) are totally unreadable ans should be checked in a further revision of the manuscript.
In attachment the authors can find additional comments to the manuscript.
In the reviewer's opinion, the present work is suitable for publication once the abovementioned points are clarified.

Round 2
Reviewer 1 Report
The authors have carefully answered to all my comments. Therefore, the paper can be move to production in its present form. The paper is a good addition to exisitng literature in the pertinent field.